# High Angiotensin-Converting Enzyme and Low Carboxypeptidase N Serum Activity Correlate with Disease Severity in COVID-19 Patients

**DOI:** 10.3390/jpm12030406

**Published:** 2022-03-05

**Authors:** Phil-Robin Tepasse, Richard Vollenberg, Nico Steinebrey, Simone König

**Affiliations:** 1Department of Medicine B for Gastroenterology, Hepatology, Endocrinology and Clinical Infectiology, University Hospital Muenster, 48149 Münster, Germany; richard.vollenberg@ukmuenster.de; 2IZKF Core Unit Proteomics, University of Münster, 48149 Münster, Germany; nico.steinebrey@uni-muenster.de (N.S.); simone.koenig@uni-muenster.de (S.K.)

**Keywords:** ACE, CPN, bradykinin, corona virus, TLC, hypertension, ACE inhibitor, COVID-19

## Abstract

(1) Background: Angiotensin-converting enzyme 2 (ACE2) is a functional receptor of SARS-CoV-2 and counter-balances ACE in the renin–angiotensin system (RAS). An imbalance of the RAS could be associated with severe COVID-19 progression. (2) Methods: Activities of serum proteases angiotensin-converting enzyme (ACE) and carboxypeptidase N (CPN) for 45 hospitalized and 26 convalescent COVID-19 patients were investigated vs. healthy controls using labeled bradykinin (DBK) degradation with and without inhibition by captopril as a read-out. Data were correlated to clinical parameters. (3) Results: DBK degradation and CPN activity were significantly reduced gender-independently in COVID-19 and returned to normal during convalescence. ACE activity was over-active in severe disease progression; product DBK1-5 was significantly increased in critically ill patients and strongly correlated with clinical heart and liver parameters. ACE inhibitors seemed to be protective, as DBK1-5 levels were normal in moderately ill patients in contrast to critically ill patients. (4) Conclusions: CPN and ACE serum activity correlated with disease severity. The RAS is affected in COVID-19, and ACE could be a therapeutic target. Further proof from dedicated studies is needed.

## 1. Introduction

Angiotensin-converting enzyme 2 (ACE2) counter-balances the renin–angiotensin system (RAS), an important endocrine, paracrine and intracrine system involved in blood pressure homeostasis [1,2]. It is also a functional receptor of severe acute respiratory syndrome coronavirus (SARS-CoV) and required for host cell entry and subsequent viral replication (Figure 1) [3]. SARS-CoV-2, the pathogenic agent responsible for the ongoing COVID-19 pandemic, seems to recognize human ACE2 even more efficiently than SARS-CoV, increasing the ability of the virus to transmit from person to person [4]. The involvement of ACE2 and the RAS in SARS-CoV-2 infection led to controversial discussions in the general public concerning a potentially increased risk for worse COVID-19 symptoms in patients taking antihypertensive medication [5,6]. A recent meta-analysis found, however, no evidence for a positive correlation of the use of ACE inhibitors (ACEIs) and angiotensin II (Ang II) receptor blockers (ARBs) with COVID-19 severity [7]. To the contrary, slight, albeit not significant, reduction in the risk of mortality was reported [7].

A number of opinion papers [5,8,9] have appeared in the recent scientific literature asking if an imbalance of the RAS may be associated with severe COVID-19 progression (see overview of scientific results in Appendix A). In particular, gender and age differences in infection, disease severity and mortality were noted, with older people and males being most vulnerable. Co-morbidities such as the metabolic syndrome were recognized as risk factors. In all of these cases, the RAS and its major enzymes ACE and ACE2 play a key role (Appendix A). Considering the inverse relationship between ACE and ACE2, the corresponding pathophysiological processes upon viral infection are yet far from clear. Higher levels of ACE2 may increase susceptibility to COVID-19 by allowing more virus into cells, but having more ACE2 could also be organ-protective [10].

ACE2 has been investigated as a potential biomarker in cardiac dysfunction [7,8], and ACE is a diagnostic marker for sarcoidosis, an inflammatory disease affecting the lungs [12]. Circulating levels of ACE and ACE2 may now also have prognostic value in monitoring COVID-19 infection [9]. We have investigated sera of COVID-19 patients for their serum ACE activity using its substrate, the neuropeptide bradykinin (BK), as a reporter substance. In the assay [11], ACE cleaves dabsylated BK (DBK, also called DBK1–9 based on the length of its sequence) at position 5–6, leading to the formation of the characteristic fragment DBK1-5, while CPN is mainly responsible for the generation of the fragment DBK1-8 [13,14,15] (Figure 2). Using this tool, we have shown that the ACE serum activity was compromised in patients with complex regional pain syndrome (CRPS), a disease where symptoms develop locally [15,16,17,18]. In a systemic disease such as COVID-19, we expected significant changes in the serum protease activity. The working hypothesis was that in COVID-19, tissue ACE2 is lowered, leading to unopposed ACE activity and high Ang II levels as well as impaired organ protection, and that signs of this activity are reflected in the serum protease activity and measurable using the neuropeptide reporter assay (NRA).

## 2. Materials and Methods

### 2.1. Participant Selection and Patient Samples

This multicenter, prospective observational cross-sectional study included 45 hospitalized patients with laboratory-confirmed SARS-CoV-2 infection (nasopharyngeal swab and test by polymerase-chain reaction), admitted to the University Hospital Münster and Marien-Hospital Steinfurt in Germany between March and June 2020. Details of medical history and laboratory data were noted. Blood from hospitalized patients (HoP) was collected during the acute phase of disease. Disease severity was defined as critical (presence of acute respiratory distress syndrome (ARDS), *n* = 22, HoPc), severe (requiring oxygen supplementation, *n* = 12, HoPs) or moderate (neither ARDS was present nor oxygen supplementation required, n = 11, (HoPm)). ARDS was diagnosed according to the Berlin definition (bilateral opacities on chest radiograph, exclusion of other causes of respiratory failure) [19]. COVID-19 patients were categorized according to their condition at the time of blood collection. One blood sample per patient was taken within the first 24 h after admission.

Additionally, 91 individuals with laboratory-confirmed SARS-CoV-2 infection who had recovered from infection were contacted for donation of convalescent plasma in our outpatient clinic. Of these, 26 persons were manually selected to match the age and gender of inpatients and a blood sample was taken. This group of patients had only moderate symptoms, and hospitalization was not necessary. In 16 patients in this group, the experimental values of interest measured in this study (NRA, see paragraph 3.2 below) had returned to normal and were detected in the same range as for healthy probands. This range had been determined for the NRA earlier in an unrelated, so far unpublished large population study. This so-called cohort of convalescent patients with normal values (CoV-NV) presented an interesting reference group and was used for data analysis in addition to healthy probands (n = 8, HCtr), because it contained twice as many subjects as the HCtr group and was more similar to the HoP cohort with respect to gender and age.

Plasma samples were obtained from each participant after they provided informed consent. The Ethics Committee of Münster University approved the current study (local ethics committee approvals AZ 2020-220-f-S and AZ 2020-210-f-S), and the procedures were in accordance with the Helsinki Declaration of 1975 as revised in 1983. Plasma samples were protected from light and frozen (−80 °C) until measurement.

### 2.2. Laboratory Measurements and Validation of the Clinical Status

Clinical laboratory assessment included complete blood count and levels of D-dimer, creatinine, urea, C-reactive protein (CRP), ferritin, procalcitonin, lactate dehydrogenase (LDH), pseudocholinesterase (PCHE), alanine aminotransferase (ALT), aspartate transferase (AST), ɣ-glutamyltransferase (ɣGT), bilirubin, creatine kinase (CK), CK-MB, and troponin T. Simplified Acute Physiology Score (SAPS) II [20] was determined on the day of laboratory measurement and used to characterize the physiological conditions of hospitalized patients.

### 2.3. Neuropeptide Reporter Assay

ACE, captopril, and EDTA were obtained from Sigma. Protease activity was measured based on the degradation of DBK. It was performed as described [11,15,16] in triplicate with slight modifications. Briefly, DBK was ordered from Peptide Specialty Laboratories (Heidelberg, Germany). Serum (3 µL) was incubated with 500 pmol DBK for 60 min at 37 °C. Degradation products DBK1-8 and DBK1-5 were separated on thin-layer chromatography (TLC) and analyzed using JustTLC software (Sweday, Sodra Sandby, Sweden). Plate-to-plate variation was minimized by normalizing the intensity of each TLC-spot to the sum of the intensity of all spots (DBK1-9, DBK1-8, DBK1-5, relative values in percent). For inhibition experiments, saturated aqueous EDTA solution (3 µL) or captopril (1 µg, suppressed all activity in tests with recombinant ACE [15]) in water, respectively, were added to the vial containing dry DBK and dried using a speedvac. Then, the assay proceeded as usual. Inhibition experiments with captopril were performed for the HoP, CoV-NV and HCtr groups. In the case of EDTA, the measurements were terminated after the analysis of 16 patient sera showed an inhibition of more than 98%.

### 2.4. Data Analysis/Statistics

Analyses were performed for the groups as described in Table 1. In order to eliminate gender bias, analyses were additionally performed only for males. For continuous variables, we report the median with the interquartile range, and values were compared using the Mann–Whitney U (Wilcoxon) test. In the case of normal distribution, *t*-tests (2-tailed) were performed. Cohen’s d for effect size was determined by running a *t*-test on the z-scores for the calculation of the mean difference. For categorical variables, we report absolute numbers and percentages, and values were compared with Chi-square tests of association or Fisher’s exact tests. A Kruskal–Wallis test was conducted to compare more than two groups. To compare subgroups, the Bonferroni correction post hoc test was performed when variance was equal (Levene’s test), and the Games-Howell test was performed when variance was different. All of the tests were two-tailed, and *p* < 0.05 was considered to indicate a statistically significant difference. All the statistical analyses were performed using SPSS (Version 26, IBM Corp., Armonk, NY, USA).

## 3. Results

### 3.1. Patient Characteristics

The proband cohort consisted of 8 healthy volunteers (3 females, age range 25–38; 5 males, age range 31–37; HCtr), 26 mostly male ambulant convalescent patients (age range 55–70; including 2 females, aged 58 and 62; CoP) and 45 hospitalized COVID-19 patients (age range 33–81, 87% males; including 6 females, age range 30–82; HoP). Sixty-two percent of the hospitalized patients had comorbidities such as hypertonia, diabetes and co-infection. For details, see Table 1 and Table 2.

### 3.2. Neuropeptide Reporter Assay Data

#### 3.2.1. Experiments without Inhibition

NRA reference values for healthy probands had been determined earlier in unrelated studies (%; DBK1-9: 18.0 ± 7.4, DBK1-8: 67.4 ± 7.3, DBK1-5: 14.6 ± 3.8; for males: DBK1-9: 18: 15.9 ± 6.2, DBK1-8: 68.8 ± 6.8, DBK1-5: 15.3 ± 3.7; unpublished). The HCtr group showed experimental values in these ranges, which was also true for more than half (62%) of the convalescent patients (the so-called CoP-NV group). The CoP-NV group was used as an additional analysis reference in this study, because it contained twice as many subjects as the HCtr group and was more similar to the HoP cohort with respect to gender and age. The remaining patients of the CoP group (CoP-DV) showed partially similar results as measured for the hospitalized patients (HoP) (Figure 3), albeit not to the same extremes. These patients were, apart from complaints such as compromised smell, mostly asymptomatic. Whether or not the abnormal NRA values hint of lingering health problems in convalescent patients remains to be investigated. Sera of convalescent patients (CoP) did not differ significantly from healthy controls (HCtr). In HCtr, but not in HoP or CoP, DBK1-8 was significantly different between males and females (*p* = 0.031).

In hospitalized patients (HoP), reduced DBK1-9 degradation, less DBK1-8 formation and partially increased DBK1-5 formation was observed (Figure 3, for exemplary data see Figure 2b). DBK1-9 and DBK1-8 for both the healthy (HCtr) and the CoP group differed significantly at the *p* < 0.001 level from that of the hospitalized patients (CoP/HoP: DBK1-9: T (69) = 4.980, d = 1.06; DBK1-8: T (69) = −4.433, d = −0.97; for males: DBK1-9: T (52) = 5.380, d = 1.323; DBK1-8: T (52) = −5.214, d = −1.296; values were normally distributed). Sera of female patients showed a similar behavior to that of males.

While no significance could be attributed to the DBK1-5 results for the HoP, CoP and HCtr groups, we did notice considerably increased, and partially even doubled, DBK1-5 values in some patients, and those were associated with critical disease progression. Significantly increased DBK1-5 values were noted for critical patients (HoPc) compared to severe (HoPs) and moderate cases (HoPc vs. HoPs: T (28) = −2.418, *p* = 0,022, d = −0.88), while DBK1-9 and DBK1-8 did not particularly differ. Interestingly, this concerned untreated hypertonic patients. In our patient cohort, only 36% of the persons with arterial hypertension took ACEIs or ARBs. None of these patients died in the hospital, while half of the 10 patients not taking this medication (49–81 y, males) had a fatal outcome. All patients with ACEI/ARB treatment had, in contrast, DBK1-5 values in the normal range. DBK1-9 and DBK1-8 were not influenced by this type of medication.

#### 3.2.2. Experiments with Inhibition

Inhibition experiments were performed for the HoP, HCtr and CoP-NV groups. DBK serum degradation was entirely inhibited by EDTA in hospitalized (HoP) and convalescent patients (CoP), demonstrating that only metallopeptidases were involved in DBK cleavage. In healthy controls (HCtr), minor residual activity (~5%) was observed [15].

Inhibition with the ACE-specific inhibitor captopril (ACE2 was not affected [21]) prevented DBK1-9 degradation by this enzyme (experimental values termed DBK1-9C). Captopril obliterated almost completely the formation of DBK1-5 in healthy probands (98-99%) and reduced DBK degradation to 10–30% [16]. A reduction in the formation of DBK1-8 was also observed, but it is known that captopril can inhibit CPN at higher concentrations (IC50 = 2.4 × 10–4 M) [22]. There is evidence that the influence of captopril on DBK degradation and DBK1-8 formation as expressed by the difference of un- and inhibited values is gender-dependent [15]. In the healthy controls (HCtr), these correlations were significant (DBK1-9Diff: *p* = 0.002, Pearson correlation (PC) 0.849; DBK1-8Diff: *p* = 0.003, PC 0.823).

The measured values for the CoP-NV subcohort of ambulant patients were in the same range as those for healthy controls. In COVID-19 inpatient sera, DBK1-5 formation was not entirely suppressed by captopril (Figure 4). The effect of captopril on the formation of DBK1-8 was remarkable; it was lowered from 50–60% to about 20–30%. When the results were sorted according to DBK1-5 (Figure 3a), some samples showed a particularly low effect of captopril, with DBK1-5C values almost reaching the DBK1-5 values detected without inhibition. Moreover, DBK1-9C results were slightly lower (~7%) than in the other patients and ~24% less DBK1-8 was formed. As visualized in Figure 3a, these results were seen in critical patients afflicted with co-morbidities and, in particular, hypertonia. As shown for the subgroup of hypertonic males in Figure 3b, the critically ill patients differed by their increased DBK1-5 values (about twice as high, without inhibition), while patients on ACEI/ARB medication exhibited inconspicuous, balanced DBK1-5 values.

No significant correlation of NRA data between hypertonic or non-hypertonic patients, those taking or not taking ACEI/ARB, HoPs or HoPm, and those with fatal or non-fatal outcome was detected. However, when co-morbidities such as diabetes, hypertonia, melanoma and co-infection were summarized as a separate category called “other disease”, DBK1-5 and DBK1-5C differed significantly in the patient cohort (HoP; DBK1-5: T (40) = −2.286, *p* = 0.028, d = −0.74; DBK1-5C: T (49) = −2.256, *p* = 0.030, d = −0.73). Those values were also significantly different for the co-infected vs. the non-co-infected patients (*p* = 0.004/0.001). DBK1-5C distinguished critical patients from the others (HoPc vs. HoPs: *p* = 0.03; vs. HoPm: *p* = 0.033). Interestingly, the least afflicted patients, those breathing normally (HoPm), differed significantly in DBK1-9Diff. Hospitalized patients (HoP) varied significantly (mostly at the *p* < 0.001 level) from recovered patients (CoP-NV) in DBK1-9, DBK1-8 and all differences to the value after inhibition including DBK1-5Diff. Significant also was the difference in subgroups (HoP vs. CoP-NV: no inhibitor, alive, HoPm, HoPs, no hypertonia, no other disease). DBK1-5 was significantly different in the comparison CoP-NV vs. subgroups HoPs (*p* = 0,002) and “no other disease” (*p* = 0.008); DBK1-5C vs. subgroups co-infection (*p* = 0.012) and HoPc (*p* = 0.034) with CoP-NV.

### 3.3. Correlations of NRA Values and Clinical Parameters

For hospitalized patients (HoP) and the CoP-NV reference group of ambulant patients, for which the inhibition experiments had been performed, several experimental values correlated with clinical parameters as shown in Table 3. DBK1-5 significantly correlated at the 0.01 level and PC > 0.4 with SAPS II (PC 0.519), troponin (0.485), bilirubin (0.578) and ɣ-GT (0.499); DBK1-5C with SAPS II (PC 0.526), ferritin (0.411), albumin (0.441) and GOT (0.562). Remarkable also was that DBK1-9C negatively correlated with troponin (−0.538) and creatinine (−0.465) and that DBK1-9Diff, the difference between the value measured without and with inhibition, visualized dependencies with ferritin (−0.542), GPT (−0.454), GOT (−0.664), LDH (−0.468), CK-MB (−0.586), CRP (−0.429) and albumin (0.541). DBK1-8 correlated with GOT (−0.403) and CK-MB (−0.429) at this level, and DBK1-8C with troponin (0.436) and creatinine (0.429). DBK1-8Diff provided even stronger correlations (ferritin: −0.476, GPT: −0.654, albumin: 0.499, CK-MB: −0.468).

## 4. Discussion

In this study, the serum activity of ACE and CPN in sera of hospitalized COVID-19 patients (HoP) was investigated using a specific NRA based on the neuropeptide BK, which is a potent vasodilator, mild diuretic, pro-inflammatory agent and involved in the mechanism of pain [22]. The assay monitored the formation of BK fragments 1-5 and 1-8, the former originating from ACE-mediated cleavage, the latter from CPN activity as confirmed by inhibition experiments with EDTA and captopril [13,14,15]. For comparison, reference values from earlier studies were available in addition to the healthy (HCtr) and convalescent probands (CoP) of the present cohort.

Few females (HoP: 6/45, CoP: 2/26) participated in the study cohort, reflecting the higher number of males seriously affected by COVID-19. Gender-based differences for DBK degradation and DBK1-8 formation were expected, and they were noted for the latter in the healthy control group (HCtr). For the patients, the data for sera of females did not vary greatly from those of males. This is hardly surprising despite the fact that ACE activity is both age- and gender-dependent [15,23], because of the small number of contributing females and their older age. The ACE activity of post-menopausal women is similar to that of age-matched men. It goes up with age in females, because estrogen downregulates ACE, and in post-pubertal children and young adults, it is lower in females than in males [23].

For the patients in the clinic (HoP), both the degradation of DBK1-9 and the formation of DBK1-8 were significantly reduced in comparison to both healthy (HCtr) and convalescent probands (CoP), and increased DBK1-5 formation was associated with severe disease progression. DBK1-5 values correlated with clinical parameters such as the mortality parameter SAPS II and heart and liver markers. Correlations of DBK1-9 and DBK1-8 with creatinine, troponin, liver parameters, and ferritin supported the association of the serum DBK degradation capacity with heart and liver impairment as well as inflammation.

CPN activity seemed to be generally reduced in patients, while that of ACE was enhanced in particularly serious cases. We observed that patients taking inhibitors (ACEI/ARB) presented a balanced ACE level close to normal, while those patients not protected in this way had a higher mortality. In fact, half of the untreated hypertonic patients, but none of the treated, died in the hospital. Although highly interesting, this observation was not significant, not least because the study was not set up to investigate this fact. Still, the finding strengthened the results of others, who also reported a reduction in the risk of mortality [7]. DBK1-9 and DBK1-8 values were not influenced by the use of anti-hypertensives.

The specific inhibition of ACE by captopril influenced the formation of both DBK1-5 and DBK1-8 [15]. In healthy persons, DBK1-9 values of ~18% were as expected (high serum degradation capacity), DBK1-5 values were ~15% (regular ACE serum activity) and DBK1-8 values were ~67%. Inhibition with captopril lowered the ACE activity, leading to higher DBK1-9 values (~80%), DBK1-5 values under 5% and DBK1-8 results of ~20%. The difference between un- and inhibited values of DBK1-5 (~10% in healthy controls) was a measure of the depleted ACE serum activity; the differences for DBK1-9 and DBK1-8 (both ~50%) contained, in addition, a contribution of CPN. Patients with co-morbidities and critical disease progression showed reduced inhibition by captopril as measured by the DBK1-5 value in addition to lower formation of DBK1-8, implicating dysregulation of the RAS. Parameters such as DBK1-5C and the difference values with and without inhibition (DBK1-9/8/5Diff) distinguished hospitalized patients (HoP) from healthy (HCtr) or convalescent persons (CoP). Interestingly, the differences of the un- and the inhibited experimental results, in particular for DBK1-9 and DBK1-8, showed stronger correlations to clinical parameters in some cases than the measurement values themselves (Table 3). Running the NRA in conjunction with specific ACE inhibition has thus proven to be a valuable strategy when studying ACE.

It appears from the present results that more enzymes than ACE and CPN may be involved in COVID-19, but they must be metalloproteases, because all activity was suppressed by EDTA. Captopril inhibition did not seem to be as efficient, and the ACE contribution as based on the DBK1-5Diff value was ~25% lower in patients (HoP) compared to healthy controls (HCtr) and the CoP-NV group. Neprylisin as another BK-degrading metallopeptidase is a possible candidate, because it has been suggested as a target to prevent organ injury in COVID-19 [24]. More detailed investigations will be necessary to clarify such contribution.

This study detected a correlation of the measured serum ACE and CPN activities with the damage of the heart muscle, blood clotting and fibrinolysis, inflammation and limited liver function in COVID-19. In particular, the ACE axis (DBK1-5 formation) was very sensitive and correlated with mortality. Our original hypothesis that overshooting ACE activity as a result of the reduced counter-regulation by ACE2 may be a part of COVID-19 pathophysiology was confirmed. The observed reduced CPN activity is likely a result of the decreased synthesis of the enzyme in the liver [25] following damage of this organ as evidenced by clinical parameters (GOT, GPT, γGT). The sketch in Figure 5 visualizes the NRA results and their suggested role in the COVID-19 pathophysiology.

## 5. Conclusions

Our study demonstrated the correlation of the activity of the serum proteases CPN and ACE with disease severity in COVID-19. CPN activity was reduced, which could be the result of the impaired liver function observed in the disease. Increased ACE activity was detected in fatally ill patients and indicated dysregulation of the RAS. The observation that none of the hospitalized patients taking ACEI or ARB died in contrast to half of the untreated hypertonic patients supported the notion that ACE might be a therapeutic target. Biswas and Kali [8] argued in their study that the reduction of deaths compared to COVID-19 patients not treated with ACEIs/ARBs could be associated with the organo-protective nature of upregulated ACE2 induced by the medication. It is worthwhile to test this hypothesis in further experimental studies as the well established hypertonia treatment would be a straightforward approach to alleviate some of the symptoms of COVID-19 infection. This study provided evidence that ACEI/ARB medication does not enhance COVID-19 symptoms as discussed in the public for some time, but that it may be, in contrast, protective.

## Figures and Tables

**Figure 1 jpm-12-00406-f001:**
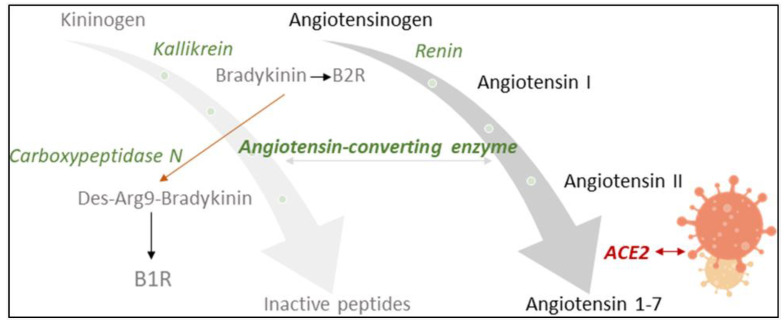
Influence of SARS-CoV2-infection on the renin–angiotensin system (RAS) and the kinin–kallikrein system, which are linked by angiotensin-converting enzyme (ACE). ACE2 counter-balances the RAS by inactivating angiotensin II. Being a functional receptor of SARS-CoV2, ACE2 availability decreases upon infection. ACE activity may subsequently increase with an impact on the cleavage of bradykinin (BK). Carboxypeptidase N (CPN) also degrades BK, changing receptor specificity from the B2 to the B1 receptor. ACE and CPN activity were measured in this study with a specially designed neuropeptide reporter assay [11].

**Figure 2 jpm-12-00406-f002:**
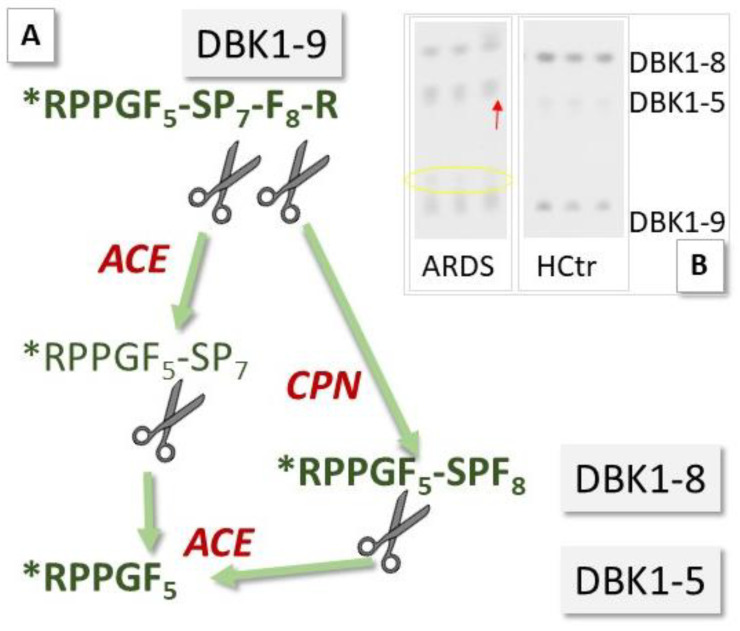
Principle of the neuropeptide reporter assay (NRA) for the study of serum protease activity. (**A**) Bradykinin (BK, sequence: RPPGFSPFR) dabsylated (see star) at the N-terminus (DBK) is degraded in serum by angiotensin-converting enzyme (ACE) and carboxypeptidase N (CPN); fragments DBK1-5 and DBK1-8 serve as readout. (**B**) Exemplary assay data show an increase in the formation of DBK1-5 and a decrease in the formation of DBK1-8 in a 66-year-old, fatally ill ARDS (acute respiratory distress syndrome) COVID-19 patient vs. data from a healthy control (female, 25 years old). The yellow circle indicates the presence of another DBK fragment, which was observed infrequently in the patient cohort and is subject to further study.

**Figure 3 jpm-12-00406-f003:**
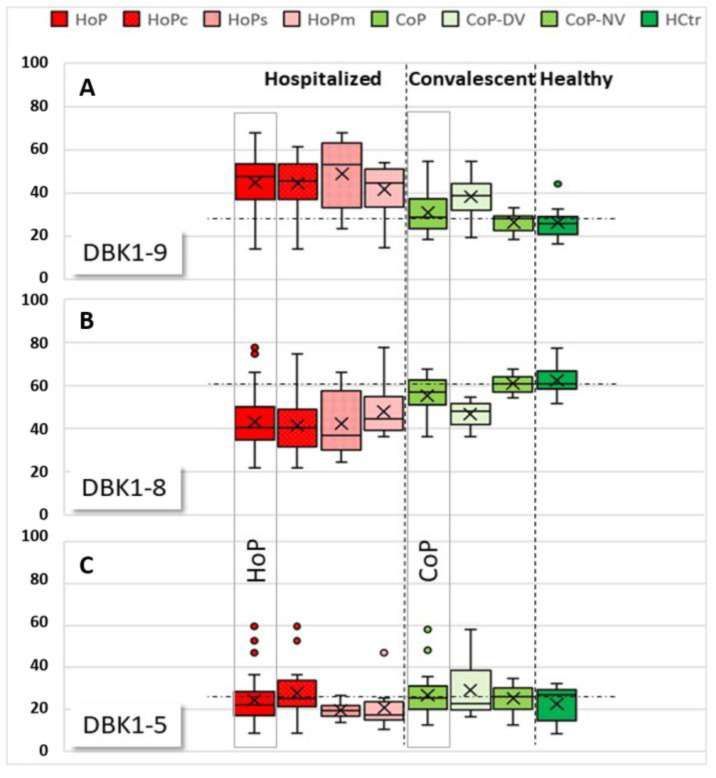
Boxplots of relative values for labeled bradykinin DBK1-9 (**A**) and its cleavage products, DBK1-8 (**B**) and DBK1-5 (**C**), for samples of the hospitalized patients (HoP) in comparison to convalescent (CoP) and healthy (HCtr) probands. Subgroups: HoPc: critical disease, HoPs: severe disease, HoPm: moderate disease, CoP-NV: CoP subgroup, where NRA results fell within the range detected in healthy probands, CoP-DV: CoP subgroup where NRA results deviated from the values measured in healthy people. The vertical dashed lines separate the groups HoP, CoP, and HCtr, the horizontal dotted lines indicate the median in the CoP-NV reference cohort for improved visualization. DBK1-9 values were higher in HoP than in HCtr and CoP-NV, indicating reduced DBK degradation and thus lower protease activity in COVID-19. DBK1-8 values were lower in HoP due to reduced CPN activity. DBK1-5 deviated to higher values in HoP and two convalescent patients (CoP-DV). DBK1-5 correlated with disease severity; values shot up in critical disease.

**Figure 4 jpm-12-00406-f004:**
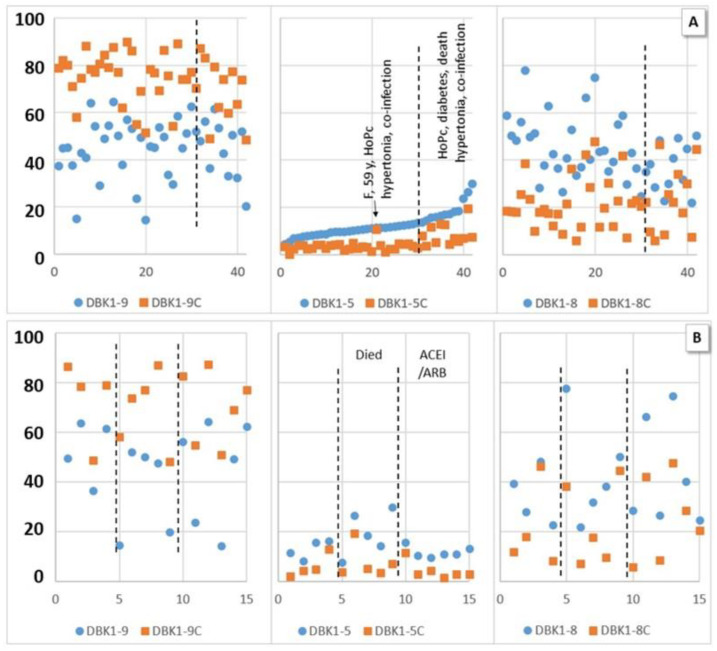
Experimental values (dabsylated bradykinin DBK1-9 and its fragments DBK1-8 and DBK1-5) in percent (y-axis; x-axis, random numbers) with inhibition by captopril (DBK1-9/8/5C, orange squares) and without inhibition (blue dots) for hospitalized patients (HoP). (**A**) Sorted by DBK1-5. High values of DBK1-5 and a lower effect of captopril were associated with other diseases in addition to COVID-19 (diabetes, hypertonia, co-infection) and poor outcome (acute respiratory distress syndrome (ARDS), death) with one outlier (female (F)). (**B**) Hypertonic male patients, data ordered by ACEI/ARB (angiotensin-converting enzyme inhibitor/angiotensin receptor blocker) medication and fatality. High DBK1-5 was observed in critical patients, while patients on ACEI/ARBs showed balanced DBK1-5 values. No effect was seen for DBK1-9 and DBK1-8.

**Figure 5 jpm-12-00406-f005:**
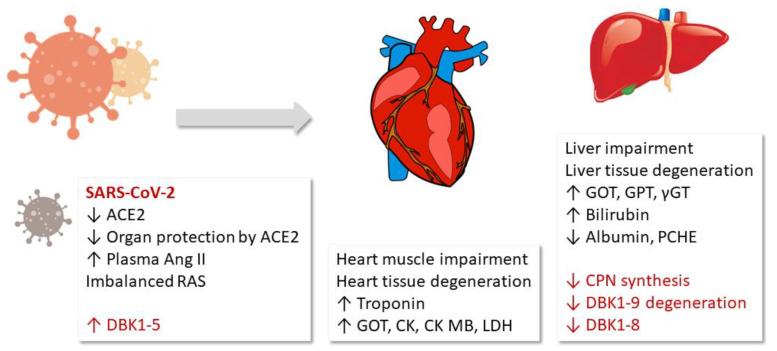
Results and hypothesis: Fragment DBK1-5 of labeled bradykinin (DBK) increased in severe COVID-19 cases due to enhanced angiotensin-converting enzyme (ACE) activity resulting from decreased ACE2 counterbalance caused by the virus. The infection led to organ damage, and clinical parameters to that effect correlated with the neuropeptide reporter assay (NRA) results. The decrease in both the serum DBK degradation and DBK1-8 formation capacity likely resulted from impaired CPN synthesis in the liver. RAS: renin–angiotensin system, LDH: lactate dehydrogenase, ɣGT: ɣ-glutamyltransferase, CK: creatine kinase, GOT: glutamate-oxalacetate transaminase, GPT: glutamate-pyruvate transaminase, PCHE: pseudocholinesterase.

**Table 1 jpm-12-00406-t001:** Cohort characteristics: differences were calculated by Kruskal–Wallis test. IQR, interquartile range; BMI, body mass index; HoP, hospitalized patients with critical (c), severe (s) and moderate (m) disease; CoP, convalescent patients; CoP-NV, convalescent patients with normal values in the NRA; HCtr, healthy controls.

Patients (Samples)		HoP				CoP		
		HoPc (*n* = 22/45)	HoPs (*n* = 12/45)	HoPm (*n* = 11/45)	*p*-Value	CoP (*n* = 26)	CoP-NV (*n* = 16/26)	HCtr (*n* = 8)
Patient Characteristics	Age, years median (IQR)	57 (49–63)	57 (49–66)	55 (38–65)	0.516	62 (55–70)	60 (55–65)	31 (25–38)
	Gender, male, n (%)	22 (100)	19 (86)	9 (82)		24 (92)	15 (94)	5 (63)
	BMI	27.3 (25–31.2)	23 (22.8–25.2)	23.8 (22.8–27.7)	**0.005**	n/a	n/a	n/a
Death (abs.)		2	7	2		0	0	0
SARS CoV-2 therapy	Cortison	9	0	2		0	0	0
Pre-existing conditions	Chronic inflammatory disease (abs.)	0	3	1		0	0	0
	Respiratory disease (abs.)	0	1	0		0	0	0
	Kidney insufficiency (abs.)	0	0	1		0	0	0
	Metastatic neoplasm (abs.)	2	1	3		0	0	0
	Diabetes (abs.)	0	3	0		0	0	0
	Arterial hypertension (abs.)	4	10	5		0	0	0
	Coronary heart disease (abs.)	1	3	2		0	0	0
Medication	Angiotensin-1 receptor antagonist (abs.)	2	1	0		0	0	0
	Angiotensin converting enzyme inhibitor (abs.)	1	2	1		0	0	0

**Table 2 jpm-12-00406-t002:** Cohort characteristics: differences were calculated by Kruskal–Wallis test. IQR, interquartile range; AST, aspartate aminotransferase; ɣ-GT, γ-glutamyltransferase; LDH, lactate dehydrogenase; PCHE, pseudocholinesterase; CK, creatine kinase; CRP, C-reactive protein; HoP, hospitalized patients with critical (c), severe (s) and moderate (m) disease; CoP, convalescent patients.

	HoP (*n* = 45)			*p*-Value	CoP (*n* = 26)
	HoPc (*n* = 22)	HoPs (*n* = 12)	HoPm (*n* = 11)		
Erythrocytes (Mio./µL)	3.4 (2.7–4.1)	4.8 (3.9–5.1)	4.7 (4.1–5.1)	*p* < 0.001	4.8 (4.6–5.2)
Hemoglobin (g/dL)	9.4 (7.9–11.6)	13.7 (11.9–15)	13.6 (11.8–14.9)	*p* < 0.001	14.6 (13.8–15.7)
Leukocytes (×10^9^/L)	9.1 (6.8–11.4)	5.3 (3.7–7)	4.1 (3.5–5.9)	*p* = 0.001	5.5 (4.96–6.9)
Lymphocytes (rel., %)	10.5 (7.2–14.1)	21.3 (13.1–22.5)	22.1 (14.6–21.1)	*p* = 0.001	28.2 (24.8–31.4)
AST (U/L)	71 (45.8–103)	41 (34–75)	38.5 (27–53.5)	*p* = 0.008	30.5 (24.3–34.8)
ALT (U/L)	42.5 (28.3–67)	34 (28–43)	25.5 (18.3–46.8)	*p* = 0.204	28.5 (24.5–34.8)
ɣ-GT (U/L)	127 (60–204)	54 (26–60)	42 (26–134.5)	*p* = 0.003	27.5 (21–50)
PCHE (U/L)	3457 (2749–5026)	6291 (4175–7998)	7695 (5806–8711)	*p* < 0.001	8956 (7948–10408)
Bilirubin (mg/dL)	0.6 (0.4–1.3)	0.5 (0.3–0.6)	0.4 (0.3–0.75)	*p* = 0.107	0.5 (0.4–0.7)
LDH (E/L)	424 (329–543)	392 (264–418)	268.5 (229.5–412)	*p* = 0.005	217.5 (194.3–239)
Creatinine (mg/dL)	1 (0.68–1.8)	0.8 (0.7–1.1)	1 (0.8–1)	*p* = 0.69	0.9 (0.8–1)
Urea (mg/dL)	19.5 (13–51)	11 (9–22)	11 (9–23.3)	*p* = 0.065	14 (13–16.5)
CK (U/L)	148.5 (57–394)	132 (46–206)	81 (59–186)	*p* = 0.515	131.5 (78.5–148)
CK-MB (U/L),	20 (16–27)	15 (8.8–17.8)	11 (5–19)	*p* = 0.012	5 (5–5)
Troponin T (ng/L)	32 (15.5–118)	12.5 (6.4–23)	8.5 (5.3–18.6)	*p* = 0.003	n.d.
D-Dimer (mg/L)	2.6 (1.9–8.7)	0.8 (0.7–1.4)	0.7 (0.4–1.5)	*p* < 0.001	n.d.
CRP (mg/dL)	14.8 (5.5–24.8)	6 (3.7–9.4)	1.8 (0.8–3.3)	*p* < 0.001	0.5 (0.5–0.5)
Ferritin (µg/L)	898 (685–1561)	692 (358–1065)	371 (197–617)	*p* < 0.001	200.5 (92–402.8)
Procalcitonin (ng/mL)	0.53 (0.16–2.04)	0.11 (0.07–0.25)	0.09 (0.06–0.13)	*p* < 0.001	0.05 (0.04–0.07)
SAPS-II Score	56 (34–73)	22 (15–32)	17 (13–24)	*p* < 0.001	n.d.

**Table 3 jpm-12-00406-t003:** Pearson correlations for bivariate correlations of NRA data with and without inhibition for hospitalized patients (HoP) and the reference group of convalescent patients (CoP-NV) with clinical parameters. ^*^ significant at the 0.05 level, ^**^ significant at the 0.01 level.

	DBK1-9	DBK1-9C	DBK1-9Diff	DBK1-8	DBK1-8C	DBK1-8Diff	DBK1-5	DBK1-5C	DBK1-5Diff
SAPS II		−0.387 ^*^	−0.339 ^*^	0.289 ^*^		−0.371 ^*^	0.519 ^**^	0.526 ^**^	
Troponin	−0.313 ^*^	−0.538 ^**^			0.436 ^**^	−0.347 ^*^	0.485 ^**^		0.373 ^*^
Erythrocytes		0.285 ^*^						−0.343	
Hb		0.296 ^*^	0.304 ^*^			0.317 ^*^		−0.337 ^*^	
Ferritin	0.346 ^*^		−0.542 ^**^	−0.392 ^**^		−0.476 ^**^		0.411 ^**^	
GPT		−0.346 ^*^	−0.454 ^**^			−0.503 ^**^	0.350 ^*^	0.317 ^*^	
GOT		−0.357 ^*^	−0.664 ^**^	−0.403 ^**^		−0.654 ^**^	0.389 ^**^	0.562 ^**^	
γGT				−0.283 ^*^		−0.338 ^*^	0.499 ^**^	0.318 ^*^	
PCHE			0.291 ^*^			0.363 ^*^	−0.343 ^*^		
Bilirubin						−0.294 ^*^	0.578 ^**^	0.302 ^*^	0.451 ^**^
Albumin	−0.299 ^*^		0.541 ^**^	0.383 ^**^		0.499 ^**^		−0.441 ^**^	
PCT		−0.340 ^*^			0.296 ^*^				
LDH	0.291 ^*^		−0.468 ^**^			−0.358 ^*^		0.280 ^*^	
CK-MB	0.384 ^**^		−0.586 ^**^	−0.429 ^**^		−0.468 ^**^		0.586 ^**^	−0.330 ^*^
CRP			−0.429 ^**^	−0.325 ^*^		−0.383 ^**^		0.451 ^**^	
Creatinine		−0.465 ^**^			0.429 ^**^				
Lymphocytes, relative	−0.287 ^*^		0.385 ^**^	0.289 ^*^				−0.371 ^**^	0.303 ^*^
Lymphocytes, absolute	−0.356 ^*^			0.282 ^*^					
D-Dimers							0.333 ^*^		
Leukocytes								0.490 ^**^	
Urea								0.317 ^*^	
CK								0.495 ^**^	

## Data Availability

All data are available on request.

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
