# Peer review of "High Angiotensin-Converting Enzyme and Low Carboxypeptidase N Serum Activity Correlate with Disease Severity in COVID-19 Patients"

_jpm, 2022, doi:10.3390/jpm12030406_

Round 1
Reviewer 1 Report
- Good study. It is important to understand the interaction between Covid and ACE and ACE2, and how variation in ACE2 / ACE activity could have prognostic influence.
Reviewer 2 Report
Thanks for the opportunity to review the manuscript by Tepasse and cols.
The manuscript shows an exciting analysis of serum ACE and CPN activities for hospitalized and convalescent COVID-19 patients compared to healthy controls using labeled DBK degradation with and without inhibition by captopril. The authors identified DBK degradation and CPN activity reduced gender-independently in COVID-19 and returned to normal during recovery. The ACE activity was over-active in severe disease progression; product DBK1-5 was significantly increased in critically ill patients and strongly correlated with clinical heart and liver parameters. ACE inhibitors seemed to be protective as DBK1-5 levels were normal in sick moderately patients compared to critically ill patients.
The manuscript looks well-written, and the study design includes patients in several severity stages; however, in my opinion, some minor issues should be attended to before its acceptance to be published.
The introduction section is extensive and vague (in some paragraphs). It should be shortened to the minimum necessary information to understand the justification and research question. Also, at the first's paragraph end, the authors state: "To the contrary, slight, albeit not significant, reduction in the risk of mortality was reported." Please include the appropriate citation.
2.1 section: "One blood sample per patient was taken within the first 24 h after admission." Please include a brief description of the number of days after the admission? Were patients' selection adjusted for the days of symptoms before admission? Please explain and include the appropriate data.
Line 111: "This group of patients had only moderate symptoms, and hospitalization was not necessary. In this group..." Please discuss the potential bias in selecting convalescent subjects since they had only moderate symptoms.
Figure 3 is unnecessary and repetitive with the previous paragraph.
Section 2.4. Data Analysis/Statistics, line 139: "analyses were additionally performed only for males."
There are two problems; previously, the authors stated that they were matched by age and gender, page 4, line 119.
Please describe clearly how many subjects were included.
Minor mistakes/typos:
Line 32, correct "Figure 1"
Section 2.3, line 143: "(also called DBK1-9 based on the length of its sequence)" is repetitive, previously has been described its meaning.
